# Comparative Compositions and Activities of Flavonoids from Nine Sanghuang Strains Based on Solid-State Fermentation and In Vitro Assays

Tian Li [1], Yuxia Mei [1,*], Ji Li [1], Wendi Yang [1], Fanfan He [1], Jiaxin Ge [1], Fei Chen [2], Yicheng Yang [1], Aowen Xie [1], Yangyang Liu [1] and Yunxiang Liang [1,*]

1   State Key Laboratory of Agricultural Microbiology, College of Life Science and Technology, Huazhong Agricultural University, Wuhan 430070, China
2   Hubei Mckesson Biotechnology Co., Ltd., Xianning 437100, China
*   Correspondence: mei@mail.hzau.edu.cn (Y.M.); liangyunxiang@mail.hzau.edu.cn or lyxhzau@gmail.com (Y.L.); Tel.: +86-27-87281040 (Y.L.)

**Abstract:** Sanghuang, a traditional Chinese medicinal herb obtained from numerous related fungal species in the genus *Sanghuangporus*, contains many bioactive substances that display a variety of beneficial pharmacological activities, including antioxidant, antitumor, and antidiabetic. We collected wild fruiting bodies from various Chinese localities, obtained nine pure sanghuang strains (termed S1 to S9), cultured the strains by solid-state fermentation, extracted and purified sanghuang flavonoids (termed SHFs) from mycelia, and analyzed their antioxidant abilities and $\alpha$-amylase inhibitory ($\alpha$-AI) activities. SHFs from strains S2, S6, S7, and S9 displayed strong DPPH radical scavenging abilities and iron reducing abilities, while SHFs from S1, S3, S5, and S8 had strong $\alpha$-AI activities. SHF components were analyzed by HPLC in combination with a Chinese medicine fingerprint similarity evaluation system and statistical analyses. SHFs from the nine strains showed high fingerprint similarity. Fifteen peaks in the chromatograms (termed 1–15) were subjected to cluster analysis, which revealed that differences in SHF composition were related to geographic origin and host species. The strains with strong antioxidant activities had relatively large peak 5 and peak 9 areas, while those with strong $\alpha$-AI activities had relatively large peak 13 areas. Such variation in SHF activities is attributable to differences in their components. Our findings indicate that careful selection of SHFs based on these activities will strengthen their potential development as antioxidant and antidiabetic agents.

**Keywords:** sanghuang; solid-state fermentation; flavonoids; antioxidant activity; $\alpha$-amylase inhibitory activity

## 1. Introduction

Sanghuang is a traditional Chinese medicinal herb popularly known as "forest gold". It is a collective term referring to numerous related fungal species in the order Hymenochaetales (division Basidiomycota), including *Sanghuangporus sanghuang*, *S. vaninii*, *S. baumii*, *S. toxicodendri*, *S. lonicericola*, and the previously misidentified *Phellinus linteus*. Sanghuang contains many bioactive substances that display a variety of beneficial pharmacological activities. The Shen Nong Ben Cao Jing ("Divine Farmer's Classic of Materia Medica"), written over 2000 years ago, reported that sanghuang stops bleeding, and alleviates diarrhea and abdominal pain [1–3]. Pharmacological studies of sanghuang in recent decades have documented immunomodulatory [4], antitumor [5–7], antioxidant [8–11], anti-inflammatory [12], and hypoglycemic [13,14] effects. Bioactive medicinal substances isolated and purified from these fungi consist mainly of polysaccharides, flavonoids, terpenoids, and phenolics [15]. Many studies have focused on polysaccharides, but few on flavonoids. Artificial cultivation, liquid-state fermentation (LSF), and solid-state fermentation (SSF) are the methods commonly used to isolate medicinal substances, since

obtaining fruiting bodies for the wild species is very difficult [16]. Artificial cultivation techniques are time-consuming, and not well developed. LSF and SSF techniques generally lead to more efficient harvesting of mycelia and production of sufficient amounts of secondary metabolites.

Flavonoids are secondary metabolites widely distributed in plants and medicinal fungi, and display a variety of pharmacological activities, including antitumor, antioxidant, anti-inflammatory, and antidiabetic [17]. Sun et al. described the advantages of flavonoids in macroscopic control in the tumor microenvironment (TME) and discussed the potential clinical significance and future research areas of flavonoids in the regulation of TAMs. These results provide promising directions for the study of antitumor drugs, while providing new ideas for the pharmaceutical industry to develop more effective forms of flavonoids [18]. Wang et al. reviewed in vitro, in vivo, and clinical models used to document the anti-inflammatory activity of flavonoids, and systematically mapped their anti-inflammatory constitutive relationships based on cross-comparison with flavanones, flavanols, and isoflavones [19]. Yan et al. compared oxidation resistance of flavonoids in mulberry fruiting bodies vs. fermentation broth. The two extracts showed moderate to high antioxidant activity, with respective $EC_{50}$ values of 4.98 and 2.39 mg/mL. The antioxidant effect of flavonoids in 0.02% fermentation broth extract was stronger than that of vitamin C [20].

The production of excessive free radicals resulting from oxidative stress is a causative factor in aging-related disorders, and many types of cancer and metabolism-related diseases. Antioxidants are often effective in ameliorating oxidative stress-induced damage. Accordingly, a major research focus in nutraceutical and cosmetic industries is the development of safe and effective natural antioxidants [21]. Antioxidant ability assays are generally classified into two main categories based on their reaction types: hydrogen atom transfer- or electron transfer-based assays. In view of economy and easiness of operation, detection methods, including 2′-Azinobis-(3-ethylbenzthiazoline-6-sulphonate) (ABTS) and 2,2-diphenyl-1-picrylhydrazyl (DPPH) radical scavenging ability assays and determination of copper or iron reduction ability, have been more widely used to assess antioxidant activity [22]. The term $\alpha$-amylase inhibitor ($\alpha$-AI; also known as "starch blocker" or "carbohydrate blocker") refers to a class of substances that display inhibitory activity against human pancreatic and salivary $\alpha$-amylases, including glycoside hydrolase inhibitors [23]. $\alpha$-AI effectively inhibits activity of these enzymes in the oral and gastrointestinal tracts and suppresses hydrolysis and digestion of carbohydrates from ingested foods; therefore, many studies have addressed its role in weight loss, hypoglycemia, and hypolipidemia. Pandey et al. evaluated antidiabetic activity of flavonoid extracts from four medicinal plant species based on their antioxidant activity (by DPPH radical scavenging method) and $\alpha$-AI activity [24]. Sanghuang, in view of its abundant flavonoid components, is considered a potential source for novel flavonoid-based drugs and health products. Tian et al. studied the efficacy of sanghuang extract and found that it showed certain antioxidant capacity [25]. Liu et al. reported that sanghuang extract had a hypoglycemic effect in a diabetic rat model [26]. However, previous research has been mostly limited to a single sanghuang strain, and there is a lack of comparative research on different strains.

In the present study, nine sanghuang strains were obtained from wild fruiting bodies, cultured by SSF, and flavonoid components were isolated and purified from mycelia. Compositions of the various flavonoid samples were analyzed by a Chinese medicine fingerprint similarity evaluation system, based on HPLC and statistical analyses. Comparative effects of the samples on DPPH radical scavenging ability, iron reducing ability, and $\alpha$-AI activity were evaluated based on in vitro assays.

## 2. Materials and Methods

### 2.1. Materials and Reagents

Nine sanghuang strains were isolated and purified from wild fruiting bodies, and subjected to fungal internal transcribed spacer (ITS) identification. A *Sanghuangporus sanghuang*

strain termed S1 was provided by the Strain Conservation Center at Huazhong Agricultural University. The eight other strains from various Chinese localities used in the study were: S2 (*S. sanghuang*; Shaanxi), S3 (*S. sanghuang*; Shennongjia), S4 (*S. sanghuang*; Shanxi), S5 (*S. sanghuang*; Tibet), S6 (*S. toxicodendri*; Shaanxi), S7 (*S. toxicodendri*; Shennongjia), S8 (*S. baumii*; Heilongjiang), and S9 (*S. vaninii*; Shandong). All strains were maintained on potato dextrose agar (PDA) slants at 4 °C.

One $cm^2$ slices were cut from the slants, inoculated onto PDA plates, and incubated at 28 °C for 25 d. Four 1 $cm^2$ slices were harvested from fresh PDA plates, inoculated into a liquid seed medium, and incubated at 28 °C for 7 d. The resulting seed solution was inoculated into LSF and SSF media (10% inoculum) and incubated at 28 °C. The seed medium consisted of: potato infusion juice 200 g/L, glucose 20 g/L, $KH_2PO_4$ 1 g/L, and $MgSO_4 \cdot 7H_2O$ 0.5 g/L. The LSF medium consisted of: mass fraction 3% soybean meal, 3% soluble starch, $KH_2PO_4$ 1 g/L, and $MgSO_4 \cdot 7H_2O$ 0.5 g/L. The SSF medium consisted of: corn grits: potato residue = 6:1, solid–liquid ratio = 5:4. Dimethyl sulfoxide (DMSO) was sourced from Aladdin Technology (Guangzhou, China). Hydrochloric acid was from Wuhan Huashun Biological Co (Wuhan, China). Potatoes, potato residue, and soybean meal were sourced from local markets. Resins (AB-8, DM130) were from Shanghai Moser Scientific Equipment Co (Shanghai, China). Other reagents were from Sinopharm Chemical Reagent Co. (Shanghai, China).

### 2.2. Optimization of Flavonoid Extraction Conditions

Mycelia were collected following 1 mo fermentation and pretreated for subsequent extraction as described by Liang et al. [16] with minor modification. Mycelial samples were extracted twice with one volume ethanol. Primary extraction conditions (temperature, ethanol volume fraction, extraction time, and material–liquid ratio) were investigated by single-factor experiments, specifically, with temperatures of 40, 50, 60, 70, 80, and 90 °C; ethanol content of 40%, 50%, 60%, 70%, 80%, and 90%; extraction times of 2, 4, 6, and 8 h; and material–liquid ratios of 1:20, 1:40, 1:60, 1:80. The extraction rate of the total flavonoids in the extraction solution was determined using the $NaNO_2$-Al $(NO_3)_3$ colorimetric method with rutin as standard. The standard curve for rutin was determined by Equation (1) below, and the flavonoid content by Equation (2). Flavonoid extraction conditions were further optimized by a four-factor, three-level orthogonal test based on single-factor experiment results. Details of orthogonal test design are shown in Table 1.

$$Y = 0.4353\ X - 0.0002, R^2 = 0.9993 \tag{1}$$

Y: absorbance of reaction solution at 510 nm; and X: rutin concentration (mg/mL);

$$\text{Flavonoid content (mg/g)} = CV/m_1 \times 100 \tag{2}$$

C: flavonoid extract concentration (mg/mL); V: extract volume (mL); and $m_1$: mycelial dry weight (g).

**Table 1.** Orthogonal test design.

| Level | Factors | | | |
|---|---|---|---|---|
| | A [1] | B [2] | C [3] | D [4] |
| 1 | 60 | 4 | 50% | 1:40 |
| 2 | 70 | 6 | 60% | 1:60 |
| 3 | 80 | 8 | 70% | 1:80 |

[1] Temperature (°C), [2] Time (h), [3] Ethanol volume fraction (%), [4] Material–liquid ratio.

### 2.3. Growth of Strains under SSF

Mycelia of the nine strains under SSF were collected, dried, and ground. Flavonoid content, mycelial yield, and total mycelial flavonoid content on days 7, 14, 21, 28, and 35 were calculated respectively by Equations (2)–(4).

$$\text{Mycelial yield (\%)} = m_2/M \tag{3}$$

$m_2$: mycelial dry weight (g) following 60-mesh sieve treatment; and M: dry weight (g) of SSF medium;

$$\text{Flavonoid content (mg)} = 100 \times Y_2 \times C_2 \tag{4}$$

$Y_2$: hyphal yield; and $C_2$: flavonoid content of hyphae (mg/g).

### 2.4. Purification of Flavonoids

Sanghuang flavonoids (SHFs) were purified using macroporous resin [27]. Suitable resin material was selected by measuring adsorption rate of resin and desorption rate of elution solution on flavonoids. For each resin, adsorption rate and desorption rate were calculated by Equations (5) and (6), and the best macroporous resin was selected. Optimal values were determined for SHF concentration, SHF pH, SHF volume, water wash volume, and ethanol elution volume. Purified SHFs were obtained under these optimized purification conditions. Flavonoid purity was calculated by Equation (7).

$$\text{Adsorption rate} = (C_3 - C_4)/ \, C_3 \times 100\% \tag{5}$$

$$\text{Desorption rate} = C_5 \times V_3 \times (C_3 - C_4) \times V_2 \times 100\% \tag{6}$$

$$\text{Flavonoid purity} = C_6 \times V_4/m_3 \tag{7}$$

$C_3$: mass concentration (mg/mL) of total flavonoids in adsorbent; $C_4$: mass concentration of total flavonoids in equilibrium solution; $C_5$: mass concentration (mg/mL) of total flavonoids in desorption solution; $V_2$: volume (mL) of adsorbent; $V_3$: volume (mL) of desorption solution; $C_6$: concentration (mg/mL) of flavonoid purification solution; $V_4$: volume (mL) of flavonoid purification solution; and $m_3$: mass (mg) of flavonoid purification solution after freeze-drying.

### 2.5. DPPH Radical Scavenging Ability of SHFs

This parameter was determined using the methods of Kamarozaman et al. [28] with minor modifications. DPPH was dissolved in anhydrous ethanol to a final concentration of 80 µg/mL. SHF samples (100 µL) at concentrations 5, 10, 25, and 50 µg/mL were mixed in 96-well plates with 100 µL DPPH solution, with SHFs in absolute ethanol as a background control, absolute ethanol with DPPH as a control, and ascorbic acid with DPPH as a positive control. Reactions were performed in the dark for 30 min at room temperature, and absorbance at 517 nm ($OD_{517}$) was measured. DPPH radical scavenging rate was calculated by Equation (8). Triplicate independent replicates were performed for each sample.

$$\text{DPPH radical scavenging rate (\%)} = 1 - [OD_{517} \, (\text{sample}) - OD_{517} \, (\text{background control})]/OD_{517} \, (\text{control}) \times 100 \tag{8}$$

### 2.6. Iron Reducing Ability of SHFs

This parameter was determined using the methods of Gebeyehu et al. [29]. Accordingly, 1 mL of sample solution, 2.5 mL of phosphate buffer (pH 6.6, 0.2 mol/L), and 2.5 mL of 1% $K_3[Fe(CN)_6]$ were mixed in a test tube, incubated in a water bath at 50 °C for 20 min, cooled rapidly, added to 2.5 mL 10% TCA, mixed well, and centrifuged at 6000 r/min for 10 min. The resulting supernatant (3 mL) was mixed with 0.6 mL 0.1% $FeCl_3$, added to 3 mL distilled water, shaken, mixed with 0.6 mL 0.1% $FeCl_3$, added to 3 mL distilled water, and

shaken again. $OD_{700}$ of reaction solution was measured with distilled water as a control. The iron reducing ability was calculated by Equation (9).

$$\text{Iron reducing ability} = OD_{700} \text{ (sample)} - OD_{700} \text{ (control)} \tag{9}$$

### 2.7. α-AI Activity

The α-AI activity was determined using the methods of Nisar et al. [30] with minor modifications. Amylase solution (2 mg/mL; 50 μL), diluted amylase inhibitor (100 μL), and PBS (200 μL) were mixed and incubated in a water bath at 37 °C for 20 min. Soluble starch (5 mg/mL; 400 μL) was added to the reaction solution and incubated at 37 °C for 10 min. DNS (250 μL) was added, incubated in a boiling water bath for 10 min, and the solution was cooled to room temperature. $OD_{520}$ of 4× diluted reaction solution was measured. Dosages of added reagents are shown in Table 2. Inhibition rates were calculated by Equation (10).

$$\text{Inhibition rate (\%)} = [(1 - (A3 - A4)/(A1 - A2)] \times 100 \tag{10}$$

A1, A2, A3, A4: $OD_{520}$ values of the blank group, blank control group, test group, and test background group, respectively.

**Table 2.** Dosage table for α-AI activity assay of SHFs.

| μL | Blank Group (A1) | Blank Control Group (A2) | Test Group (A3) | Test Background Group (A4) |
|---|---|---|---|---|
| Amylase solution | 50 | 0 | 50 | 0 |
| SHFs | 0 | 0 | 100 | 100 |
| PBS | 300 | 350 | 200 | 250 |
| Soluble starch | 400 | 400 | 400 | 400 |
| DNS | 250 | 250 | 250 | 250 |

### 2.8. Composition Analysis of SHFs

Purified SHF samples (Section 2.4) were dissolved in acetonitrile (chromatographic grade) (final concentration of 1 mg/mL), shaken, and filtered through a microporous membrane. The resulting sample solution (1 mL) was placed in an injection vial. HPLC analysis was performed using a ZORBAX Eclipse Plus C18 column (4.6 mm × 250 mm; 5 μm) (Agilent, Santa Clara, CA, USA) and a SPD-20A (V) UV detector (Shimadzu, Kyoto, Japan) with the following conditions: mobile phase formic acid/ water (A) and acetonitrile (B); flow rate 1.0 mL/min at 40 °C; injection volume 10 μL; detection wavelength 355 nm; elution program 0–5 min, 90% A, 10% B; 5–30 min, 90–60% A, 10–40% B; 30–31 min, 60–90% A, 40–10% B; and 31–40 min, 90% A, 10% B.

### 2.9. Statistical Analysis

Experimental data were expressed as mean ± SEM, and differences between means were analyzed using the statistical software program GraphPad Prism 8.0 (San Diego, CA, USA). Datasets involving more than two groups were analyzed by a one-way analysis of variance (ANOVA) followed by Tukey's multiple comparisons test. Differences with $p < 0.05$ or $p < 0.01$ were regarded respectively as significant or highly significant.

## 3. Results

### 3.1. The Effects of Various Factors on the Extraction of Total Sanghuang Flavonoids (SHFs)

The effects of temperature, ethanol volume fraction, extraction time, and material–liquid ratio on the extraction of total SHFs were evaluated. The SHF content increased as temperature increased from 40 to 70 °C, then declined at higher temperatures (Figure 1a). Zulkifli et al. [31] had similar results, indicating that there is an optimal temperature range for leaching of flavonoids. SHF extraction efficiency was higher for ethanol volume fraction 60% than for values of 40%, 50%, 70%, 80%, and 90% (Figure 1b). Xia et al. reported similar findings for extraction of Larix gmelinii flavonoids, indicating the existence of

an optimal extractant (ethanol) volume fraction [32]. The total SHF content increased as extraction time increased from 2 to 4 h, with a maximal value at 4 h, then declined with longer times (Figure 1c). The result indicates that the extraction time was too long to be conducive to flavonoid extraction. SHF content was higher for a material–liquid ratio 1:40 than for values of 1:20, 1:60, and 1:80 (Figure 1d). Liu et al. obtained similar results in the optimization of the material–liquid ratio for extraction of flavonoids from the exocarp of three genera of coconuts [33]. Evidently, an adequate degree of contact between solid material and liquid promotes substance exchange and extraction of total flavonoids. The optimal material–liquid ratio in the present study was 1:40, in terms of both extraction efficiency and economy. Through the optimization of extraction conditions, experimental materials and labor could be minimized, which helped to keep costs down.

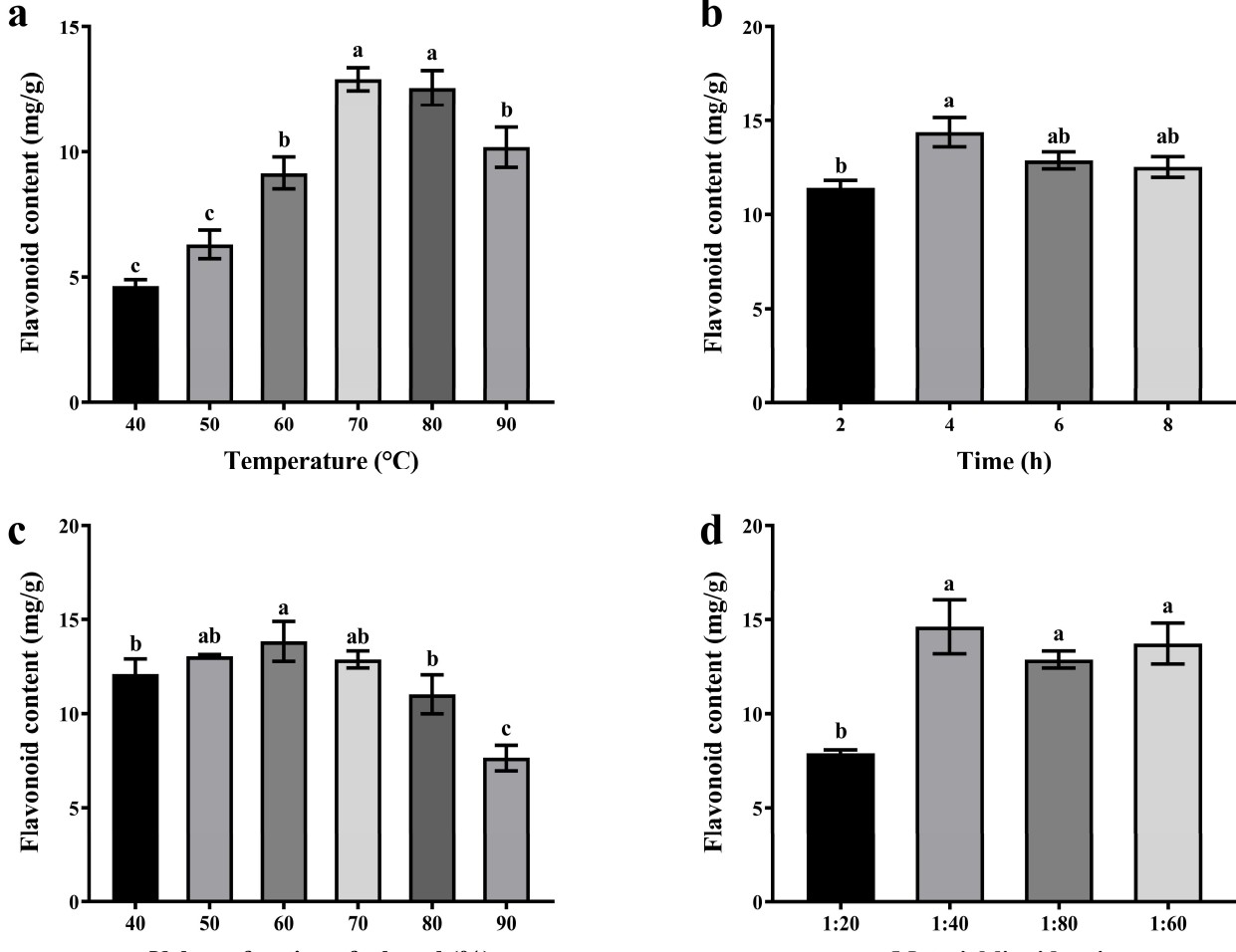

**Figure 1.** The effects of various factors on extraction of total SHFs from mycelia: (**a**) temperature (°C); (**b**) time (h); (**c**) ethanol volume fraction (%); and (**d**) material–liquid ratio (*w/v*). Differing letters above bars indicate significant ($p < 0.05$) differences according to Tukey's multiple comparisons test.

### 3.2. Optimization of the Total SHF Extraction Program Based on the Orthogonal Test

A four-factor, three-level L9 ($3^4$) orthogonal test for the optimization of the total SHF extraction program was designed and performed. Results are shown in Table 3. Among the nine experiments, No. 7 (A3B1C3D2; i.e., 80 °C, 50%, 4 h, 1:40) showed the highest SHF yield. Ranges of these four factors were 5.224 (A), 0.859 (B), 2.612 (C), and 1.118 (D), indicating the order A > C > D > B; i.e., temperature had the greatest effect on the SHF extraction process, followed by extraction time, material–liquid ratio, and ethanol volume fraction.

**Table 3.** Orthogonal test results.

| Experiment Number | A | B | C | D | SHF Content (mg/g) |
|---|---|---|---|---|---|
| 1 | 1 | 1 | 1 | 1 | 9.11 |
| 2 | 1 | 2 | 2 | 2 | 10.56 |
| 3 | 1 | 3 | 3 | 3 | 11.93 |
| 4 | 2 | 1 | 2 | 3 | 13.39 |
| 5 | 2 | 2 | 3 | 1 | 13.69 |
| 6 | 2 | 3 | 1 | 2 | 11.79 |
| 7 | 3 | 1 | 3 | 2 | 18.06 |
| 8 | 3 | 2 | 1 | 3 | 14.94 |
| 9 | 3 | 3 | 2 | 1 | 14.26 |
| K1 [1] | 31.600 | 40.564 | 35.846 | 37.063 | |
| K2 [1] | 38.873 | 39.195 | 38.216 | 40.418 | |
| K3 [1] | 47.272 | 37.987 | 43.683 | 40.266 | |
| $\overline{K}1$ [2] | 10.533 | 13.521 | 11.949 | 12.354 | |
| $\overline{K}2$ [2] | 12.958 | 13.065 | 12.739 | 13.472 | |
| $\overline{K}3$ [2] | 15.757 | 12.662 | 14.561 | 13.422 | |
| Range | 5.224 | 0.859 | 2.612 | 1.118 | |

[1] Sum of SHF content for each factor at each level. [2] Average of sum of SHF content for each factor at each level.

### 3.3. SSF of Nine Sanghuang Strains

Mycelia from nine sanghuang strains as described in Section 2.1 were harvested following SSF (5–35 day culture period). Mycelial yield, flavonoid content per g of mycelium, and total SHF content per 100 g of SSF medium were measured and compared (Figure 2). Mycelial yield was highest for strains S6 and S7 (Figure 2a), flavonoid content per g of mycelium was highest for S3 and S8 (Figure 2b), and total SHF content was highest for S3 and S5 (Figure 2c). For each strain, a maximal value of total SHF content was used for comparative analysis, which indicated that content was highest for S3 and S5, and these values were significantly different from those for the other seven strains (Figure 2d).

Based on results of their 2001 study, Robinson et al. concluded that SSF, relative to other techniques, yields more stable products in smaller fermenters, requires less energy, and allows simpler downstream processing measures [34]. Gustavo et al. used the logical equation and the Luedeking–Piret equation to explain the higher enzyme yields obtained in SSF relative to immersion fermentation (SmF). Productivity was higher for an SSF system than for an SmF system in all three enzyme production scenarios studied [35]. Similarly, in the present study, high sanghuang hyphae amounts and flavonoid contents were obtained by SSF that were sufficiently adequate for our subsequent experiments. SSF is expected to become a reliable way to cultivate large rare fungi and obtain their effective secondary metabolites.

### 3.4. Screening of Macroporous Resins and Optimization of Purification Conditions

Adsorption of SHFs was evaluated using four macroporous resins (D101, AB-8, DM301, and polyamide resin). Each of these resins had a pale white color and opaque globules. D101 and AB-8 are nonpolar or weakly polar, whereas DM301 and polyamide resin are more strongly polar. Adsorption and desorption rates were calculated and are shown in Table 4. The order of adsorption rates was polyamide resin (99.04 ± 0.58%) > DM301 (62.03 ± 0.55%) > D101 (57.71 ± 0.27%) > AB-8 (54.06 ± 0.68%). The order of desorption rates was D101 (90.38 ± 0.40%) > DM301 (88.81 ± 0.13%) > AB-8 (76.96 ± 0.98%) > polyamide resin (2.22 ± 0.04%). On the basis of these findings, DM301 was selected as the optimal resin for subsequent purification. Macroporous resin has stable physical and chemical properties, high mechanical strength, simple purification process operation, acid and alkali resistance, organic solvent resistance, and strong applicability to industrial production, which makes up for the shortcomings of traditional mechanical separation technology and membrane separation technology. Actually, macroporous resins have been used for a long time in the purification of flavonoids. Ismail et al., in a study of baobab (Adansonia digitata fruit pulp)

flavonoids, performed static adsorption and desorption tests of four macroporous resins, and concluded that HPD-500 resin was the most suitable for separation and purification of flavonoids [36]. In an analogous study of sunflower flavonoids, Ri et al. determined that AB-8, among eight macroporous resins, was the most suitable for purification [37]. The findings of these two studies differ from those of the present study. Evidently, the effects of various resins differ depending on the flavonoid samples being studied, i.e., selection of suitable purification material should be made on a case-by-case basis, depending on the study samples.

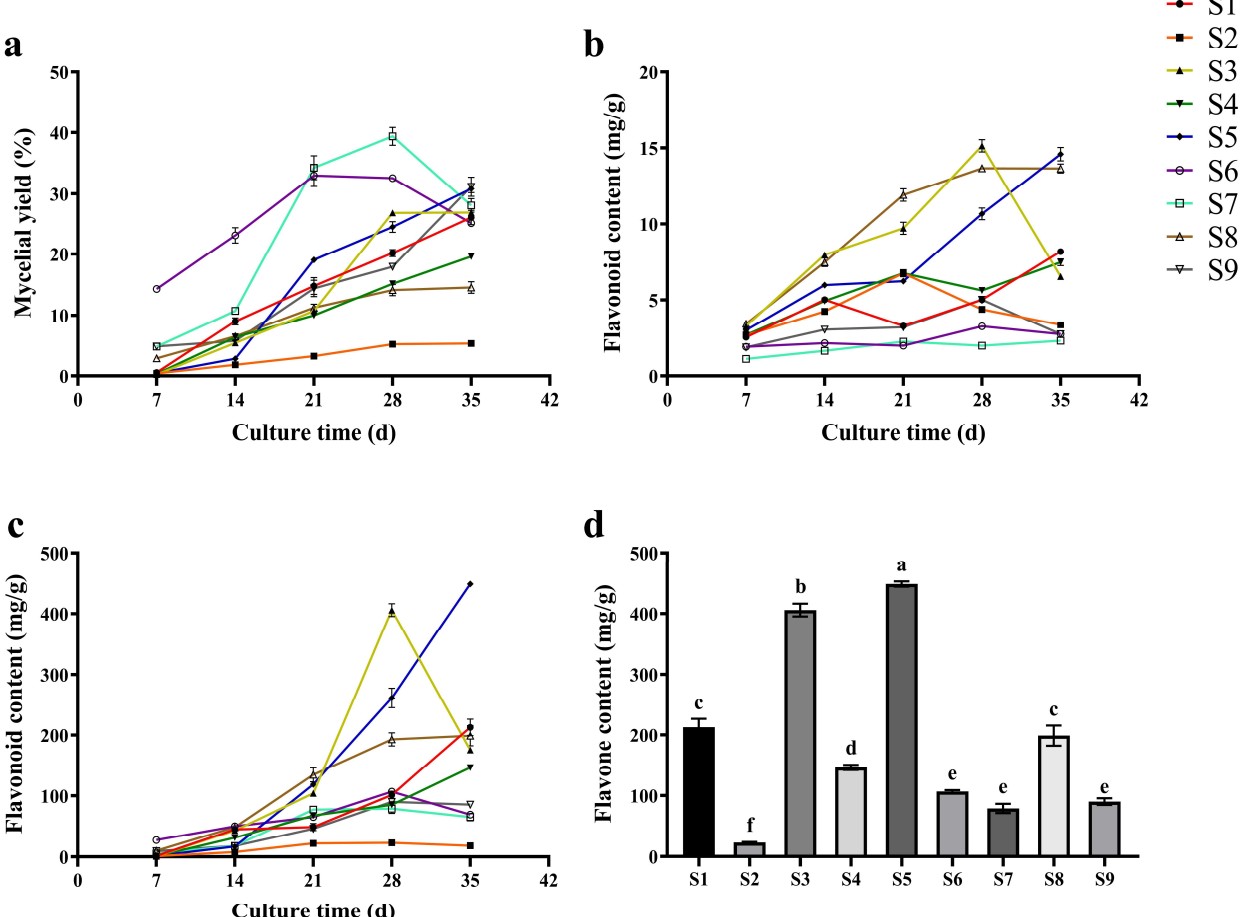

**Figure 2.** Mycelial growth and SHF production under SSF: (**a**) mycelial yield under SSF; (**b**) SHF content per g of mycelia under SSF; (**c**) total SHF content under SSF; and (**d**) the maximum of total SHF content accumulation in nine strains. Differing letters above bars indicate significant ($p < 0.05$) differences according to Tukey's multiple comparisons test.

**Table 4.** Adsorption rates and desorption rates of SHFs in various macroporous resins.

| Macroporous Resin | Adsorption Rate (%) | Desorption Rate (%) |
|---|---|---|
| D101 | 57.71 ± 0.27 [c] | 90.38 ± 0.40 [a] |
| AB-8 | 54.06 ± 0.68 [d] | 76.96 ± 0.98 [c] |
| DM301 | 62.03 ± 0.55 [b] | 88.81 ± 0.13 [b] |
| Polyamide resin | 99.04 ± 0.58 [a] | 2.22 ± 0.04 [d] |

Differing letters above bars indicate significant ($p < 0.05$) differences according to Tukey's multiple comparisons test.

We investigated SHF separation conditions and the effects of various factors in adsorption chromatography. The effects of loading solution concentration, pH, and volume on DM301 adsorption rate are shown in Figure 3a–c, and the determinations of water

wash volume and ethanol elution volume based on flavonoid concentration are shown in Figure 3d,e.

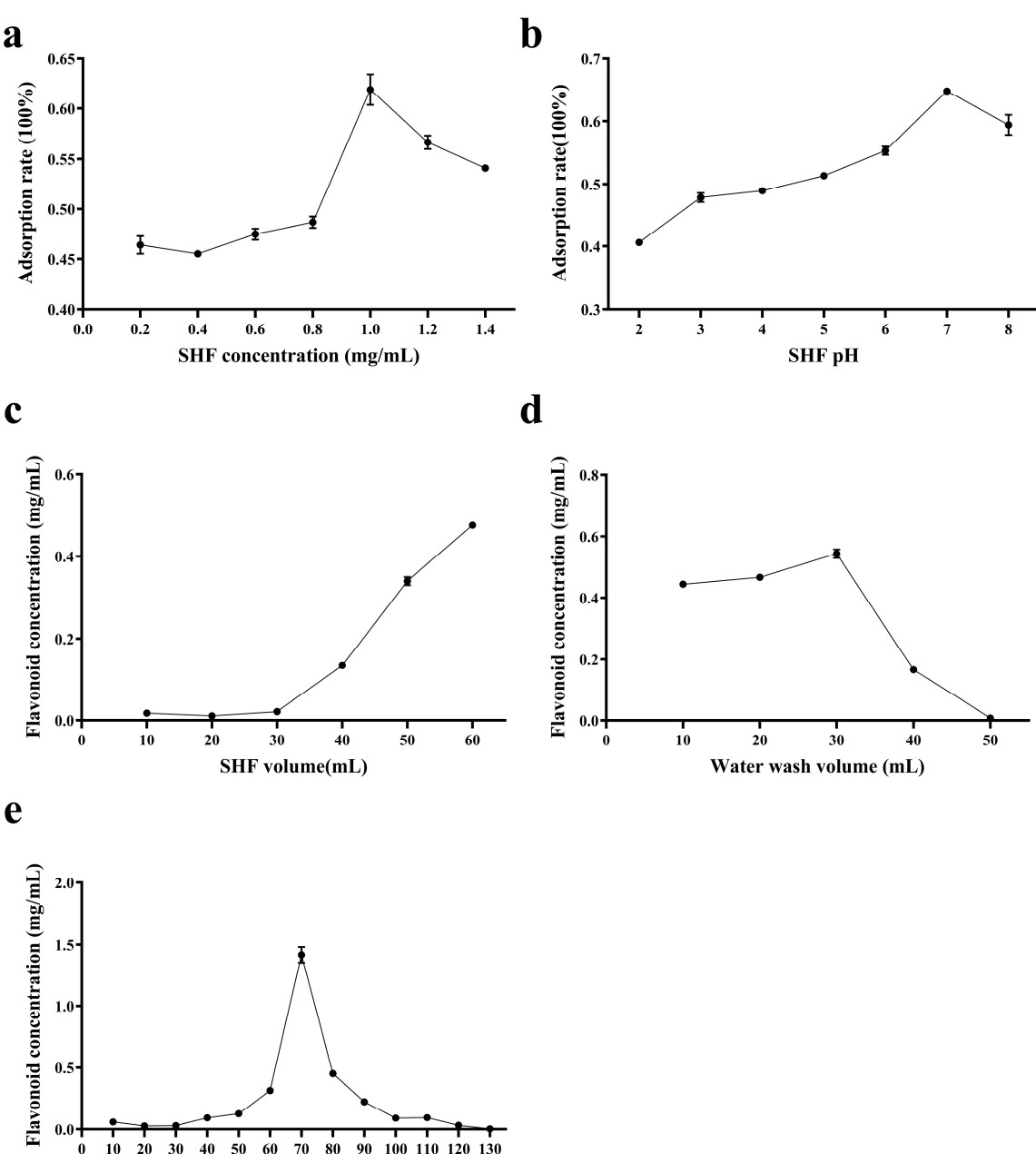

**Figure 3.** The effects of various factors on SHF purification: (**a**) SHF concentration; (**b**) SHF pH; (**c**) SHF volume; (**d**) water wash volume; and (**e**) ethanol elution volume.

Excessive sample concentration (volume) leads to flavonoid leakage, and consequent sample waste and column bed blockage, whereas small loading concentration (volume) results in low adsorption efficiency of resin. In our experiments, the adsorption rate was highest at neutral pH (7). Under acidic conditions, flavonoids generate salts, which interfere with their adsorption. Under alkaline conditions, the dissociation of flavonoids into hydrogen ions and corresponding cations inhibits hydrogen bond interactions and thereby reduces adsorption rate. At washing volume 50 mL, flavonoid content was close to zero, indicating that flavonoids were tightly bound and essentially all impurities were eluted. At ethanol elution volume 120 mL, there were essentially no flavonoids left behind,

indicating that all flavonoids were collected. Based on our experiments, the optimized conditions were: SHF concentration 1 mg/mL; SHF pH 7; SHF volume 35 mL; water wash volume 50 mL; and ethanol elution volume 120 mL. Under these conditions, the final purity of SHFs was 75.17%.

In the above-mentioned studies by Ri et al., parameters such as loading concentration, loading volume, and flow rate were optimized following the purification of the sunflower flavonoids, resulting in respective 83.5% increases in purity [37]. Macroporous resins are clearly effective for flavonoid purification, result in high purity, and will be useful for future functional studies.

### 3.5. Comparative Antioxidant Activities and α-AI Activities of Various SHFs

Data on DPPH radical scavenging ability and iron reducing ability are presented in Figures 4 and 5. For all SHFs, DPPH radical scavenging ability was positively correlated with concentration. At low concentrations, DPPH radical scavenging ability and iron reducing ability of ascorbic acid were significantly higher than those of SHFs. At high concentrations, these abilities of ascorbic acid were close to those of SHFs. DPPH radical scavenging ability was stronger for S2, S6, S7, and S9 than for other strains. Iron reducing ability was concentration-dependent for all SHFs, and was again stronger for S2, S6, S7, and S9 than for other strains. In conclusion, SHFs from S2, S6, S7, and S9 had the strongest antioxidant activity, and are the most suitable for development of antioxidant agents.

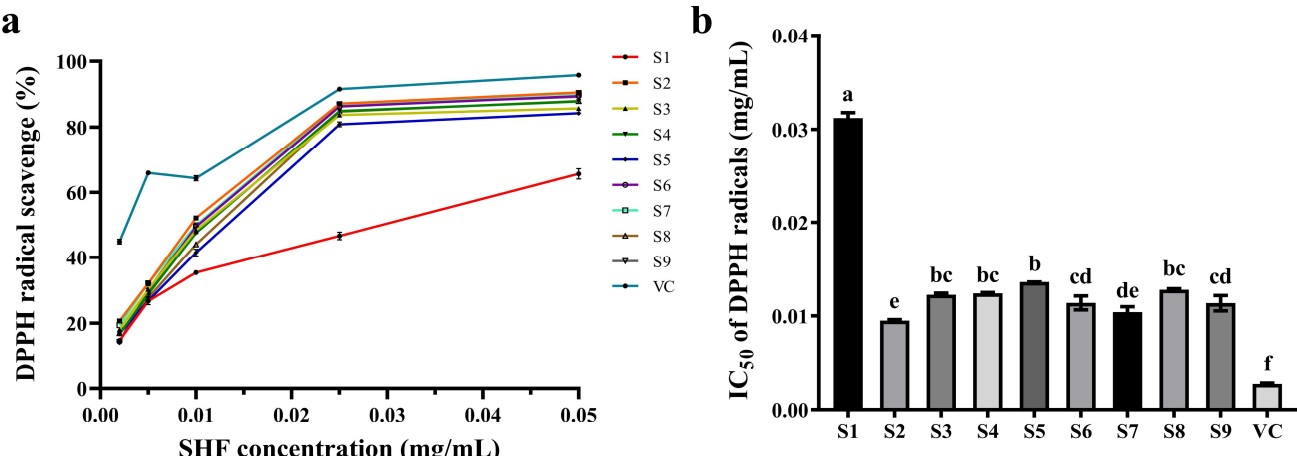

**Figure 4.** DPPH radical scavenging abilities of nine SHFs: (**a**) DPPH radical scavenging abilities of nine SHFs within concentration range 0.002–0.05 mg/mL; and (**b**) $IC_{50}$ for DPPH radicals. Differing letters above bars indicate significant ($p < 0.05$) differences according to Tukey's multiple comparisons test.

Flavonoid extracts from fermented products of *P. igniarius* were shown by Wang et al. to be promising sources of antioxidant-related food additives [20]. Flavonoid extracts from fermented products are designated "Generally recognized as safe" (GRAS), and have good potential for application in health foods. In a study by Lin et al. comparing antioxidant activities of ethanol extracts from three sanghuang species, the DPPH radical scavenging ability of *S. sanghuang* was significantly ($p < 0.05$) higher than those of the other two [38]. This observation differs from our experimental findings, reflecting individual variation in antioxidant functions of sanghuang species. Sanghuang antioxidant activities are evidently determined by environmental factors—not solely by species. The selection of sanghuang strains should not only consider the variety, but also combine the comprehensive analysis of other factors, such as the growth environment of sanghuang.

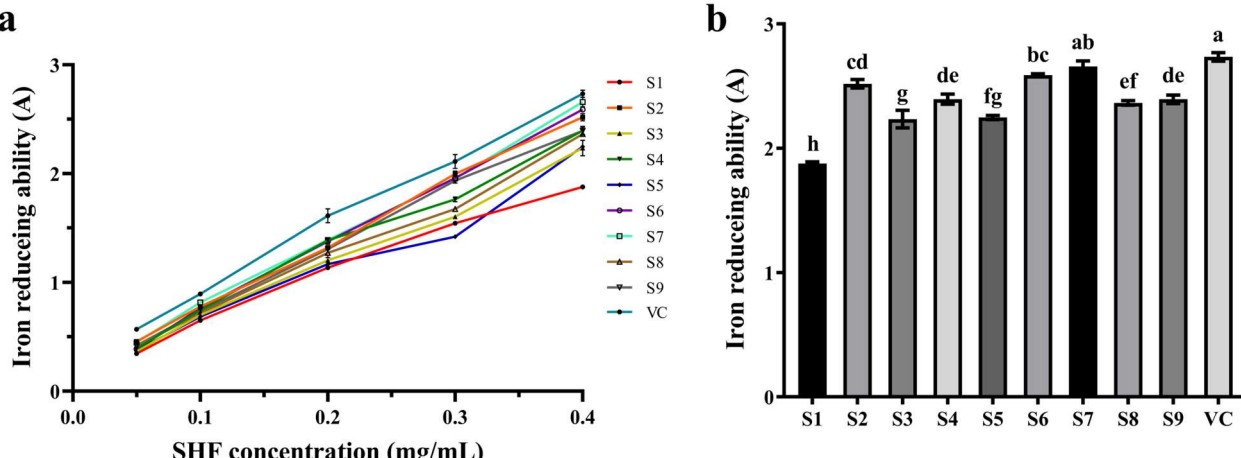

**Figure 5.** Iron reducing abilities of nine SHFs: (**a**) iron reducing abilities of nine SHFs within concentrations range 0.05–0.4 mg/mL; and (**b**) iron reducing ability of flavonoids at a concentration of 0.4 mg/mL. Differing letters above bars indicate significant ($p < 0.05$) differences according to Tukey's multiple comparisons test.

Comparative data on $\alpha$-AI activities of SHFs from various strains are presented in Figure 6a,b. Under SSF, $\alpha$-AI activity of produced flavonoids and acarbose was strongest for strain S1 at low concentrations, and for strain S5 at high concentrations. Overall, $\alpha$-AI activities of SSF-produced flavonoids were stronger for strains S1, S3, S5, and S8, and weaker for strains S2, S4, S6, and S7.

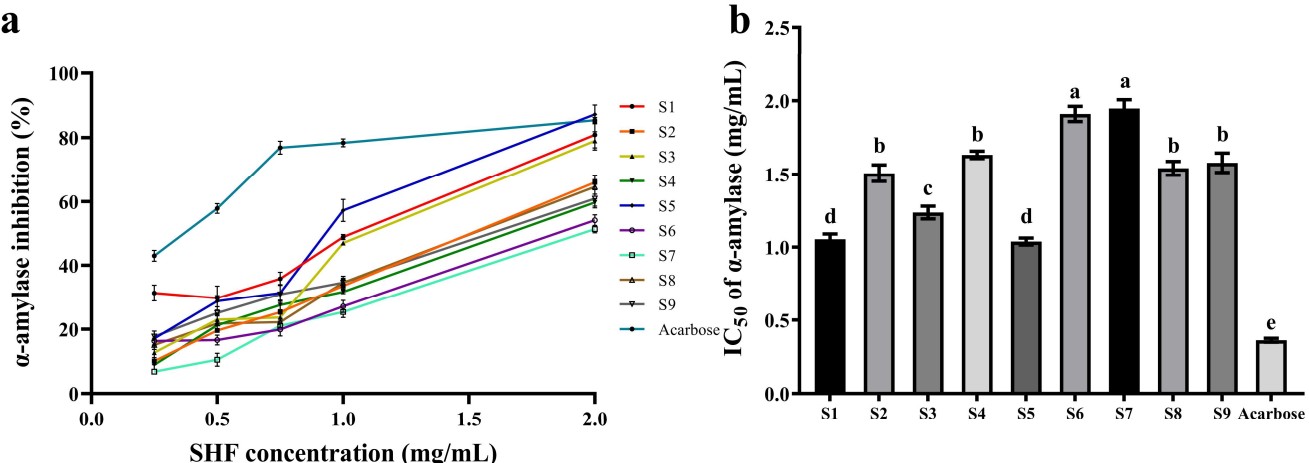

**Figure 6.** $\alpha$-AI activities of nine SHFs: (**a**) $\alpha$-AI activities of nine SHFs within concentration range 0.25–2 mg/mL; and (**b**) $IC_{50}$ of $\alpha$-amylase. Differing letters above bars indicate significant ($p < 0.05$) differences according to Tukey's multiple comparisons test.

Antidiabetic activities of fungal and plant extracts are often evaluated on the basis of $\alpha$-AI activities. In a study by Pandey et al. comparing antidiabetic activities of flavonoids from extracts of four medicinal plants, $\alpha$-AI activity was highest for the fern *Aleuritopteris bicolor* ($IC_{50}$ 651.58 $\pm$ 10.32 µg/mL) [24]. In the present study, $\alpha$-AI activities of flavonoids from various sanghuang strains were analyzed in vitro, and the data provide a basis for predicting their antidiabetic potential.

### 3.6. Component Analysis of Mycelial SHFs from Nine Sanghuang Strains

High performance liquid chromatography fingerprint analysis has been widely used in study of Chinese medicinal material composition [39,40]. In this study, we used HPLC

fingerprint analyses to compare the components of flavonoids. HPLC fingerprint analyses of the nine SHFs was performed (Figure 7), and the results were evaluated using the Chinese medicine fingerprint similarity evaluation system (Table 5). Similarity values were high: 0.60, 0.90, 0.89, 0.88, 0.79, 0.94, 0.91, 0.91, and 0.97 for strains S1–S9, respectively. Cluster analysis, a method for analyzing and simplifying datasets, is useful for highlighting similarities and differences among data points [41]. Peak areas from the chromatograms in Figure 7 were subjected to cluster analysis using the OriginPro analysis tool. Results (Figure 8) led to clustering of strains into three groups: group I (S2 and S6), group II (S1, S2, S3, S4, and S5), and group III (S7, S8, and S9). Strains S2 and S6 in group I were from Shaanxi (see Section 2.1). Group II strains were from parasitic fruiting bodies on mulberry trees, while group III strains were from parasitic fruiting bodies on lacquer, tyrannical horse butternut, or poplar trees. Differential SHF composition among strains was evidently associated with geographic origin and with host species.

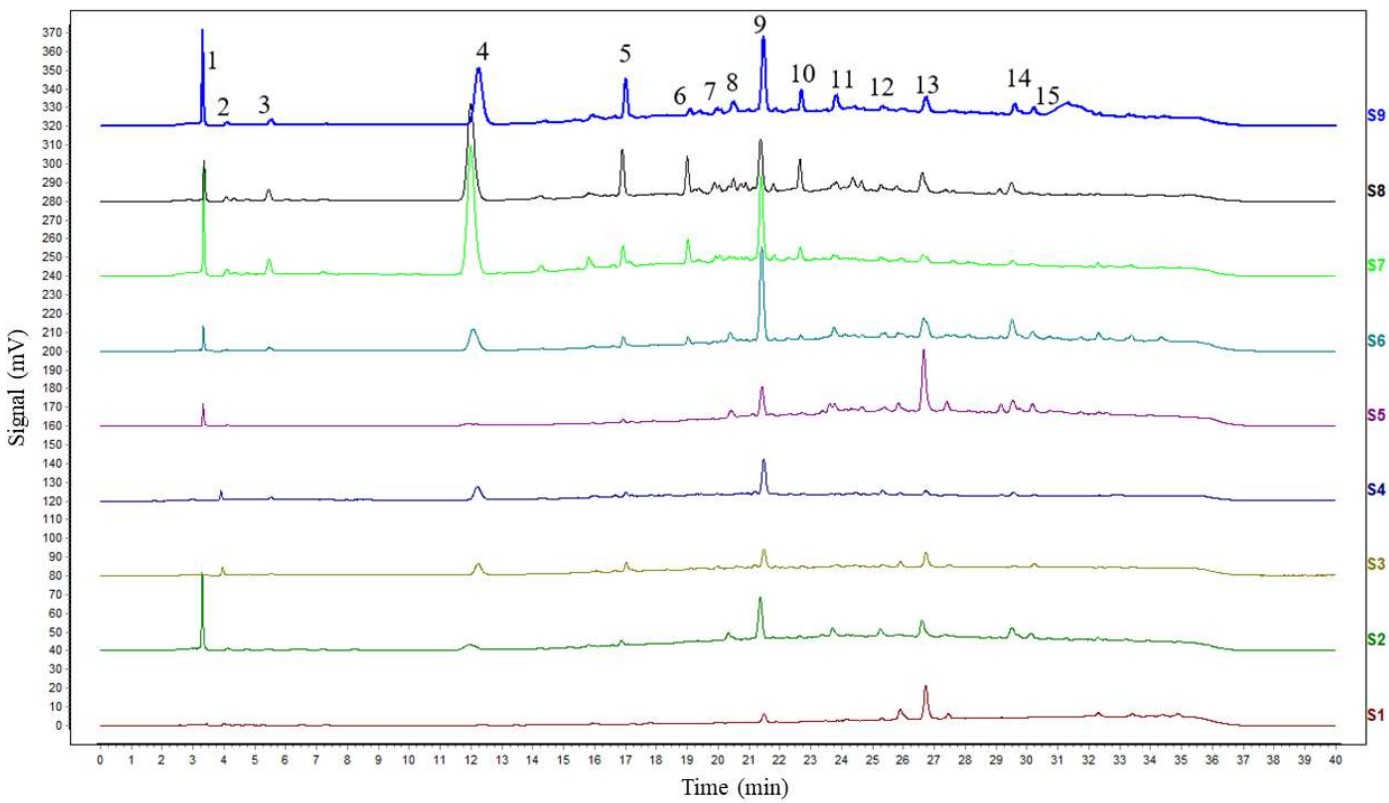

**Figure 7.** HPLC fingerprint analysis of SHFs from strains S1–S9.

**Table 5.** Component similarities of SHFs from the nine strains under SSF.

| Strain | Similarity |
|--------|------------|
| S1 | 0.60 |
| S2 | 0.90 |
| S3 | 0.89 |
| S4 | 0.88 |
| S5 | 0.79 |
| S6 | 0.94 |
| S7 | 0.91 |
| S8 | 0.91 |
| S9 | 0.97 |

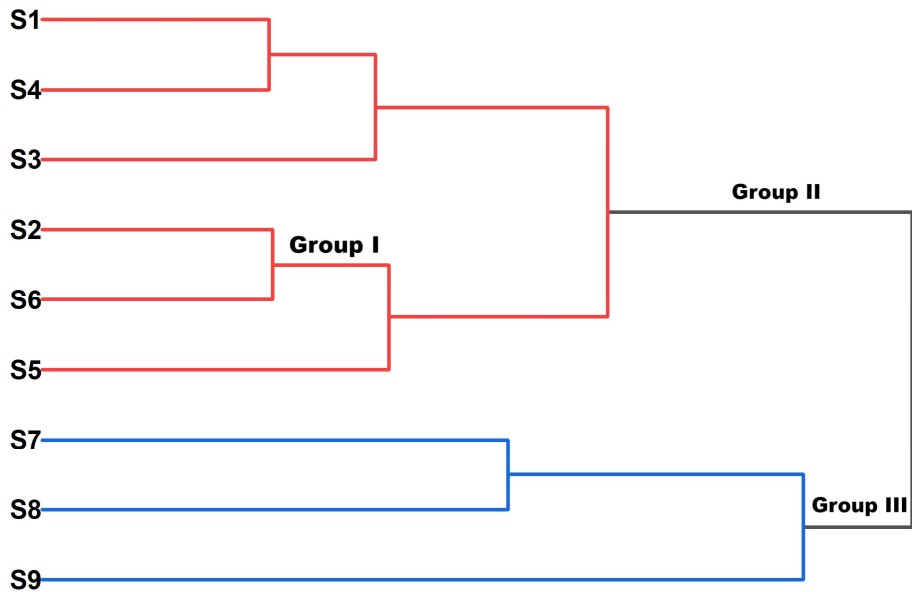

**Figure 8.** Cluster analysis of SHFs from the nine strains.

HPLC fingerprint analyses in combination with SHF activity analyses revealed that strains characterized by strong antioxidant activity (S2, S6, S7, and S9) had relatively large peak 5 and 9 areas, while strains with strong $\alpha$-AI activity (S1, S3, S5, and S8) had relatively large peak 13 areas. Such variation in SHF activities is attributable to differences in their components. Xiao et al. found that anticoccidial activities of Tiefeng samples were closely related to their major active components and to natural conditions of the source regions [42]. In a study by Nijat et al., 21 batches of Rosa rugosa, samples were purchased from different Chinese herbal medicine markets in several provinces of China, and the authors concluded that common components in alcohol extracts had free radical scavenging abilities, and that the chemical composition of samples was similar but the content was significantly different, which could explain the disparity of the antioxidant activity of per sample [43]. In the present study, different sanghuang strains showed differential flavonoid composition and functional strength. For sanghuang, best-performing strains should be selected based on differential needs of functional development.

## 4. Conclusions

Nine sanghuang strains were separated from fruiting bodies, cultured by SSF, and their flavonoid (SHF) contents and mycelial yields were measured. SHFs were isolated and purified by solvent extraction and adsorption chromatography under optimized conditions, and their compositions and activities were compared. HPLC fingerprint and clustering analyses showed that differential SHF composition among the strains was related to geographic origin and to host species. SHFs displayed strong antioxidant activity and $\alpha$-AI activity in in vitro assay, but these activities varied among the strains. Our findings indicate that careful selection of SHFs based on these activities will strengthen their potential development as antioxidant and antidiabetic agents.

**Author Contributions:** T.L.: methodology and writing—original draft. Y.M.: writing—review and editing, supervision, and funding acquisition. J.L.: drawing and typesetting. W.Y.: modifying the details of the article. F.H., J.G. and F.C.: investigation. Y.L. (Yangyang Liu), Y.Y. and A.X.: software. Y.L. (Yunxiang Liang): writing—review and editing and supervision. All authors have read and agreed to the published version of the manuscript.

**Funding:** This study was supported by the National Natural Science Foundation of China (grant No. 42271056).

**Institutional Review Board Statement:** Not applicable.

**Informed Consent Statement:** Not applicable.

**Data Availability Statement:** All data generated or analyzed during this study are included in this article.

**Conflicts of Interest:** The authors declare that they have no competing financial interest or personal relationships that could potentially influence the studies or findings described in this paper.

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
