# Peer review of "Comparative Compositions and Activities of Flavonoids from Nine Sanghuang Strains Based on Solid-State Fermentation and In Vitro Assays"

_fermentation, doi:10.3390/fermentation9030308_

Round 1

Reviewer 1 Report

In this paper, the characteristics of nine Sanghuang, species are observed, the antioxidant activity of the extracts is measured, and their usefulness is described. It is considered interesting to try to obtain the characteristics of each species from the various experiments. However, there are several areas for improvement, and we urge the authors to consider and describe the following.

1. Why did you choose to measure AI activity and DPPH activity? The reason is lacking. Please provide more details.

2. Fig.4 is difficult to understand. This figure is difficult to understand the concentration dependence of each sample. Can we improve the way it is shown, or show it as a line graph, etc.? There seems to be no concentration dependence for some samples, but is this due to experimental error?

3. There are two Fig.3s. Is this a misprint?

4.The resolution of the chart of HPLC analysis is low. Also, is it possible to guess the components for each peak? Also, could the difference be due to different concentrations of the extracts, or were the measurements performed with the same concentrations of extracts?

5.In the discussion section, I feel that I am only stating the results. I would like to see more concrete descriptions of the scientific findings and the results that can be expected from these results.

Author Response

Reviewer 1#:

In this paper, the characteristics of nine Sanghuang species are observed, the antioxidant activity of the extracts is measured, and their usefulness is described. It is considered interesting to try to obtain the characteristics of each species from the various experiments. However, there are several areas for improvement, and we urge the authors to consider and describe the following.

R: Thank you for your comments.

  1. Why did you choose to measure AI activity and DPPH activity? The reason is lacking. Please provide more details.

R: Thank you very much for this constructive suggestion. In the Sec. “Introduction”, we strengthened the in-depth analysis on the significance of DPPH free radical scavenging and AI activity assays according to previous studies. Antioxidants are generally effective in relieving damage caused by oxidative stress. Antioxidant ability assays are generally classified into two main categories based on their reaction types: hydrogen atom transfer- or electron transfer- based assays. In view of economical efficiency and easiness of operation, the latter detection methods, including 2'-Azinobis-(3-ethylbenzthiazoline-6-sulphonate) (ABTS) and 2,2-diphenyl-1-picrylhydrazyl (DPPH) free radical scavenging ability assays and determination of copper reduction ability or iron reduction ability, are more widely used to assess antioxidant activity [1]. Alpha-amylase inhibitor (α-AI) effectively inhibits the activity of α-amylases in the oral and gastrointestinal tract, and inhibits the hydrolysis and digestion of carbohydrates from ingested foods. Therefore, α-AI plays an important role in in weight loss, hypoglycemia, and hypolipidemia. It has also been reported in previous studies that Sanghuang showed strong antioxidant and hypoglycemic effects. e. g. Tian et al. found Sanghuang extract displayed certain antioxidant ability [2]. In the previous study of our group, Sanghuang extract has been verified to have strong hypoglycemic effect in a diabetic rat model [3]. However, the previous researches were mostly limited to a single Sanghuang strain, and there is a lack of comparative research on different strains. The aim of this study is to compare the abovementioned activities and composition of different varieties, thus contributing to deeply understand the difference in efficacy of different species.

References:

  1. Udani, J.; Tan, O.; Molina, J. Systematic review and meta-analysis of a proprietary alpha-amylase inhibitor from white bean (Phaseolus vulgaris L.) on weight and fat loss in humans. Foods 2018, 7, 63.
  2. Tian, X.M.; Dai, Y.C.; Song, A.R.; Xu, K.; Ng, L.T. Optimization of liquid fermentation medium for production of Inonotus sanghuang (Higher Basidiomycetes) mycelia and evaluation of their mycochemical contents and antioxidant activities. Int J Med Mushrooms 2015, 17, 681-691.
  3. Liu, Y.Y.; Wang, C.R.; Li, J.S.; Mei, Y.X.; Liang, Y.X. Hypoglycemic and hypolipidemic effects of Phellinus Linteus mycelial extract from solid-state culture in a rat model of type 2 diabetes. Nutrients 2019, 11.
  4. Fig.4 is difficult to understand. This figure is difficult to understand the concentration dependence of each sample. Can we improve the way it is shown, or show it as a line graph, etc.? There seems to be no concentration dependence for some samples, but is this due to experimental error?

R: Thank you for this suggestion. We've adjusted Figure 4 by replacing the bar chart with a line chart, thus it could be shown the trend of efficacy over concentration more visually. Overall, efficacy increases as the concentration increases within the set concentration range. A few samples tend to be irregular at low concentrations, which might be due to experimental errors. According to your suggestion, Figure 5 has also been revised in the same way now.

  1. There are two Fig.3s. Is this a misprint?

R: Sorry for carelessness. A redundant one has been deleted in the revised version.

4.The resolution of the chart of HPLC analysis is low. Also, is it possible to guess the components for each peak? Also, could the difference be due to different concentrations of the extracts, or were the measurements performed with the same concentrations of extracts?

R: Thank you. A chart of HPLC analysis with higher resolution has been used now. Through construction of HPLC fingerprint, the composition of flavonoids from different Sanghuang strains based on the solid-state fermentation were compared in this study. According to the principle of HPLC, the peaks at the same retention time represent the same substances separated and analyzed by a same program. The larger the peak area, the higher the content of the substance. We will identify the components of the main peaks and carry out more detailed functional studies in subsequent research. In addition, all extracts were obtained using the same procedure of isolation and purification, and their concentrations were adjusted to be consistent for HPLC analysis which was performed with three replicates, and the experiments showed good reproducibility.

5.In the discussion section, I feel that I am only stating the results. I would like to see more concrete descriptions of the scientific findings and the results that can be expected from these results.

R: Thanks for your suggestions. We have expanded discussion with a more specific and in-depth description of the results and more research literature have been cited. See sections of 3.1-3.6 for details.

Reviewer 2 Report

The manuscript contains good information and can be considered after the following changes:

Abbreviation PDA should be reserved for potato dextrose agar rather than glucose agar.

Table 4 add statistical analysis

Figure 8 should have labels to demonstrate in a better way

For DPPH, and amylase inhibition, please present each sample at different concentrations at one place so that trend can be observed. Moreover, IC50 values were calculated??

Author Response

Reviewer 2#:

The manuscript contains good information and can be considered after the following changes:

  1. Abbreviation PDA should be reserved for potato dextrose agar rather than glucose agar.

R: Done.

  1. Table 4 add statistical analysis

R: Done.

  1. Figure 8 should have labels to demonstrate in a better way

R: Thanks for your suggestion. An updated Figure 8 with a more detailed and aesthetically pleasing presentation has been employed in the revised version.

  1. For DPPH, and amylase inhibition, please present each sample at different concentrations at one place so that trend can be observed. Moreover, IC50 values were calculated??

R: Thank you for this suggestion. We've adjusted the figure by replacing the bar chart with a line chart, thus it could be shown the trend of efficacy over concentration more visually. And IC50 values have been calculated and supplemented in Figure 4b and 6b.

Round 2

Reviewer 1 Report

Dear authors

I confirm that the points I pointed out have been changed. There is no problem with this manuscript.